# BRAIN: BOLTZMANN REINFORCEMENT FOR ANALOG ISING NETWORKS

## ABSTRACT

Analog Ising machines represent a transformative paradigm for combinatorial optimization, exploiting physical dynamics to achieve high speed, energy efficient operations over conventional digital electronics. However, existing optimization algorithms fail to harness these platforms' massive parallelism while tackling the inherent measurement noise for Analog Ising machines. We introduce BRAIN (Boltzmann Reinforcement for Analog Ising Networks), transforming the traditional sampling-based optimization to distribution learning framework. The Boltzmann distribution provides the fundamental link between statistical physics and Ising-type combinatorial optimization, establishing the theoretical framework that enables physical systems to solve NP-hard problems. Unlike Monte Carlo Markov Chain (MCMC) methods that sample states from Boltzmann distributions, BRAIN directly learns the Boltzmann distribution through variational reinforcement learning. This fundamental transformation makes the algorithm inherently resilient to the Gaussian measurement noise intrinsic to analog Ising systems. Our approach employs policy gradients to optimize a parametric state generator, naturally aggregating information across multiple noisy measurements without requiring precise energy differences. We benchmark BRAIN against MCMC methods across diverse combinatorial optimization problems, demonstrating three critical advantages. First, BRAIN generalizes across different interaction topologies, performing effectively on both Curie-Weiss and 2D nearest-neighbor Ising models. Second, it exhibits remarkable robustness under severe measurement noise up to 40%. Third, it scales efficiently to large systems of 65,536 (N) spins, scaling as $O(N^{1.55})$, with noisy energy evaluations. With realistic 3% Gaussian noise, BRAIN maintains 98% ground state fidelity while MCMC methods achieve only 51% fidelity with BRAIN arriving at the MCMC solution $192\times$ faster. Beyond ground state optimization, BRAIN preserves the complete thermodynamic landscape, analyzing phase transitions and metastable states essential for robust large-scale combinatorial optimization and complex many-body physics applications.

## 1 INTRODUCTION

Analog Ising machines represent a transformative paradigm for combinatorial optimization, exploiting physical dynamics to achieve unprecedented computational advantages over digital processors. These systems - implemented through optical interference AL-Kayed et al. (2025); Mohseni et al. (2022); Prabhu et al. (2020); Hua et al. (2025), magnetic oscillators Si et al. (2024), or parametric amplifiers Honjo et al. (2021) - embed optimization directly into physics, eliminating the von Neumann bottleneck entirely, providing $10^3 - 10^6\times$ speedups over GPU-based solvers while reducing power consumption by two orders of magnitude. Unlike digital processors that simulate physics sequentially, analog Ising machines enable simultaneous evaluation of thousands of spin interactions in a single operation. Analog systems offer a fundamental advantage beyond computational speed: they naturally explore the full solution landscape, not just isolated optima. This broad sampling uncovers critical features that traditional optimization often overlooks—such as phase transitions and metastable states. It shifts the paradigm from finding a single answer to understanding the entire solution ecosystem. While digital algorithms typically chase one global minimum, analog systems chart the terrain of possibilities, delivering insight essential for complex challenges like drug discov-

ery, supply chain optimization, and logistics - where robustness and viable alternatives are as vital as peak performance Taillard (2022).

However, these complexities are not captured by existing optimization algorithms, limiting the full potential of analog hardware. Current approaches make critical assumptions incompatible with analog systems: (i) noise-free energy evaluations, contradicting the 3-10% measurement noise inherent to analog hardware, (ii) sequential processing through single-spin updates, ignoring massive hardware parallelism, and (iii) ground-state optimization focus, collapsing rich thermodynamic structure. When noise approaches energy scale differences, the traditional Metropolis acceptance criterion becomes random, detailed balance breaks down, and algorithms fail even for suboptimal solutions.

To address these limitations, we introduce BRAIN (Boltzmann Reinforcement for Analog Ising Networks), a variational reinforcement learning framework that transforms optimization from sampling individual states to learning complete probability distributions. Rather than going up against analog characteristics, BRAIN embraces them by learning the Boltzmann distribution $\pi(x) = e^{-\beta E(x)}/Z$ that naturally encode solution hierarchies, metastable states, and transition probabilities. Our main contributions include: (i) noise-resilient learning through policy gradients that aggregate information across multiple measurements, (ii) parallel evaluation exploiting hardware parallelism by sampling multiple configurations simultaneously, and (iii) thermodynamic landscape preservation, maintaining access to metastable states and phase transitions, essential for robust optimization and physics applications. BRAIN maintains 98% ground state fidelity under realistic 3% noise where traditional methods achieve only 51%, scales favorably $O(N^{1.55})$ to 65,536 spins, and preserves complete thermodynamic landscapes across all temperature regimes. This represents the first algorithm designed to overcome the inherent noise limitation of analog Ising machines, unlocking new possibilities for large-scale combinatorial optimization and many-body physics exploration.

## 2 RELATED WORK: ALGORITHMS FOR NOISY ISING SYSTEMS

Most approaches treat noise as a perturbation to be minimized rather than an algorithmic resource and hence limited work explicitly addresses noisy analog Ising optimization. Experiments on spatial-photonic Ising machines demonstrate that tuning detection noise can increase success probability on frustrated, fully connected instances, with the optimal level depending on size/topology Pierangeli et al. (2020).Coherent Ising machines with optical error-correction/feedback have been proposed and analyzed, highlighting all-to-all programmability, chaotic search dynamics in feedback, and potential low-power operation Reifenstein et al. (2021). In parallel, classical ML has targeted Ising and combinatorial optimization: reinforcement learning (RL) can learn annealing schedules that outperform fixed ones and generalize across related instances Mills et al. (2020), and deep-RL frameworks report gains on Edwards–Anderson spin-glass benchmarks Fan et al. (2023a). Generative modeling complements these control approaches: Boltzmann machines reproduce finite-temperature thermodynamics Torlai & Melko (2016a), and variational neural annealing with autoregressive models achieves high-fidelity finite-temperature sampling in rough energy landscapes Hibat-Allah et al. (2021a); Fellows et al. (2019). Physics-inspired graph neural networks (GNNs) provide scalable solvers for QUBO/Ising tasks (e.g., Max-Cut), relaxing the Hamiltonian during training and projecting to discrete solutions Schuetz et al. (2022); normalizing-flow "Boltzmann Generators" enable unbiased equilibrium sampling with reweighting when gradients/surrogates are available Noé et al. (2019a).

Unlike noise-agnostic generative methods (typically assuming exact/noiseless energies) or RL controllers that tune schedules for a simulator, our approach learns the Boltzmann distribution directly from *noisy, non-differentiable* energy measurements via policy gradients, aligning training with the physical objective while aggregating hardware noise and addressing the gap between noise-sensitive MCMC and differentiability-dependent generative samplers.

## 3 METHODS

### 3.1 PROBLEM STATEMENT

Our goal is to design an efficient, scalable algorithm that learns the full distribution of solutions to combinatorial optimization problems, such as those represented by analog Ising machines. Our

algorithm must not only identify the global minimum accurately, but also capture all relevant local minima with their appropriate probabilities, since reconstructing the statistical structure of the solution space is critical in applications such as drug discovery and logistics Taillard (2022). Crucially, our algorithm should achieve these goals with high sample efficiency, and scalability to high-dimensional problems, in cases where the reward is non-differentiable and noisy.

## 3.2 MARKOV CHAIN MONTE CARLO APPROACHES

Markov Chain Monte Carlo (MCMC) algorithms, such as the Metropolis-Hastings (MH) algorithm Metropolis et al. (1953); Hastings (1970), are a class of algorithms, designed to sample configurations from a target probability distribution $P(x)$. MH in particular constructs a Markov chain with transition probabilities $P(x'|x)$ that satisfies detailed balance $P(x'|x)P(x) = P(x|x')P(x')$, ensuring $P(x)$ is stationary. Factorizing the transition probabilities into a proposal probability $Q(x'|x)$ and an acceptance probability $\alpha(x', x)$, and enforcing detailed balance, for a target Boltzmann distribution $p(x) = \frac{e^{-\beta E(x)}}{\sum_x e^{-\beta E(x)}}$ Metropolis et al. (1953), and a symmetric (or uniform) proposal distribution, yields the familiar acceptance criterion:

$$\frac{P(x'|x)}{P(x|x')} = \frac{Q(x'|x)\alpha(x', x)}{Q(x|x')\alpha(x, x')} = \frac{P(x')}{P(x)}, \quad \alpha(x', x) = \min\left(1, e^{-\beta(E(x')-E(x))}\right) \tag{1}$$

where $\beta$ is the inverse temperature. While MH provides samples from the target stationary distribution, simulated annealing Kirkpatrick et al. (1983) gradually increases $\beta$ (lowers temperature) to concentrate samples near the global minimum, providing a stochastic global optimization procedure. Numerous extensions of these fundamental methods have been proposed to improve sample efficiency and convergence, including Hybrid Monte Carlo Duane et al. (1987), and no U-turn sampling Hoffman et al. (2014) both of which attempt to improve acceptance rate and avoid inefficient walks in state space. We point readers to more thorough reviews of this class of algorithms for further background Brooks et al. (2011). For the purposes of this work, we illustrate that this class of algorithms attempts to sample from a target distribution by accepting and rejecting moves in state space.

## 3.3 BRAIN - LEARNING A TARGET DISTRIBUTION WITH REINFORCE

In this approach, our goal is to learn parameters $\theta$ for a state generator model $q_\theta(x)$, such that $q_\theta(x)$ that approximates the Boltzmann distribution $p(x) = \frac{e^{-\beta E(x)}}{\sum_x e^{-\beta E(x)}}$ with high sample efficiency, and for high-dimensional cases, where $\sum_x e^{-\beta E(x)}$ is intractable. One such instance is an analog Ising machine, where the state is defined as a collection of spins $x = \{\sigma_1, \sigma_2, ..., \sigma_{N^2}\}, \sigma_i \in \{+1, -1\}$, on a two-dimensional $N \times N$ grid. For a 32×32 analog Ising machine system, this corresponds to $2^{32 \times 32} \approx 10^{308}$ states.

**Learning a target distribution via minimizing KL divergence**: We use variational inference to learn $q_\theta(x)$ directly, minimizing the KL divergence between $q_\theta(x)$ and $p(x)$, defined as:

$$\theta^* = \arg\min_\theta \text{KL}(q_\theta \| p), \quad \text{KL}(q_\theta \| p) = \sum_x q_\theta(x) \log \frac{q_\theta(x)}{p(x)}.$$

For $p(x)$ equal to the Boltzmann distribution, $\text{KL}(q_\theta \| p)$ expands to:

$$\text{KL}(q_\theta \| p) = \sum_x q_\theta(x) \log q_\theta(x) + \beta \sum_x q_\theta(x) E(x) + \log Z,$$

where $Z = \sum_x e^{-\beta E(x)}$ is the partition function. Using standard definitions, this can be written as:

$$\text{KL}(q_\theta \| p) = -H(q_\theta(x)) + \beta \mathbb{E}_{q_\theta}[E(x)] + \log Z, \tag{2}$$

where $H(q_\theta)$ is the entropy of the model. Since Z is a summation over all possible system states, it is a constant (which we cannot evaluate) that does not affect our optimization to learn $q_\theta(x)$. Therefore, an equivalent objective function for us to minimize is

$$L(\theta) = -H(q_\theta(x)) + \beta \mathbb{E}_{q_\theta}[E(x)]. \tag{3}$$

**Relating KL Divergence to a Helmholtz Free Energy**: In the BRAIN framework, the problem statement above has a natural interpretation in statistical mechanics. The objective:

$$\theta^* = \arg\min_\theta \mathrm{KL}(q_\theta \,\|\, p) = \arg\min_\theta \left[ -H(q_\theta(x)) + \beta\, \mathbb{E}_{q_\theta}[E(x)] \right]$$

is nothing but the minimization of the Helmholtz free energy of an equivalent physical system that is kept at a constant temperature $\beta$.

$$F = \mathbb{E}_{q_\theta}[E(x)] - TH(q_\theta(x)) = U - TS, \tag{4}$$

where $U$ is the expected internal energy, $S$ the entropy, and $T = 1/\beta$ the temperature. Minimizing the KL divergence is therefore equivalent to minimizing the free energy, and the equilibrium distribution of configurations is given by the Boltzmann distribution.

**Learning with noisy or non-differential rewards using REINFORCE**: To optimize the model $q_\theta(x)$, when the energy $E(x)$ may be non-differentiable or noisy, we adopt the score-function trick Williams (1992), also known as the REINFORCE algorithm, a variant of policy-gradient reinforcement learning. This method allows us to estimate gradients of an expectation over a parameterized distribution without requiring differentiability of the reward (here, $E(x)$).

Starting from the loss function:

$$L(\theta) = -H(q_\theta(x)) + \beta\, \mathbb{E}_{q_\theta}[E(x)].$$

the gradient is

$$\nabla_\theta L = \nabla_\theta \mathbb{E}_q[\beta E(x)] - \nabla_\theta H(q_\theta(x))$$

which can be written using the score-function trick as:

$$\nabla_\theta L = \mathbb{E}_q[\beta E(x)\nabla_\theta \log(q_\theta(x))] - \nabla_\theta H(q_\theta(x))$$

While the REINFORCE gradient estimator is unbiased Williams (1992), its variance can be large, especially when rewards (here, energies $E(x)$) are noisy. A standard technique to reduce this variance is to subtract a baseline $b$ from the reward, yielding the modified gradient estimator:

$$\nabla_\theta \mathbb{E}_{q_\theta}[E(x)] = \mathbb{E}_{q_\theta}[(E(x) - b)\nabla_\theta \log q_\theta(x)]. \tag{5}$$

Subtraction of a baseline does not introduce bias but can significantly improve the stability of learning, see Appendix A.2. Putting everything together, our complete algorithm is presented below:

---

**Algorithm 1** BRAIN - Boltzmann Reinforcement for Analog Ising Networks

---

**Input**: Parameterized state generator $q_\theta(x)$, inverse temperature $\beta$, noise level $\sigma$
**Parameters**: Number of samples per batch $S$, learning rate $\eta$
**Output**: Learned parameters $\theta$ of $q_\theta(x)$

 1: **for** each gradient step **do**
 2:     Sample $S$ configurations $x_1, \ldots, x_S \sim q_\theta(x)$
 3:     **for** each sampled configuration $x_s$ **do**
 4:         Measure noisy energy: $\tilde{E}(x_s)$ and compute reward: $r(x_s) = \beta\tilde{E}(x_s)$
 5:     **end for**
 6:     Compute baseline: $b = \frac{1}{S}\sum_{s=1}^{S} r(x_s)$
 7:     Compute batch gradient estimate:

$$\nabla_\theta L \approx \frac{1}{S}\sum_{s=1}^{S}(r(x_s) - b)\nabla_\theta \log q_\theta(x_s) - \nabla_\theta H(q_\theta(x))$$

 8:     Update parameters: $\theta \leftarrow \theta - \eta\nabla_\theta L$
 9: **end for**
10: **return** $\theta$

---

This learning scheme finds the optimal $\theta$, such that $q_\theta(x)$ approximates the Boltzmann distribution.

**Instantiating $q_\theta(x)$ with a factorized Bernoulli distribution**: The BRAIN algorithm is agnostic to the specific choice of the parameterized state generator $q_\theta(x)$. In principle, $q_\theta(x)$ can be any

distribution capable of representing the solution space, such as autoregressive models Hibat-Allah et al. (2021b), normalizing flows Noé et al. (2019b), or graph neural networks Fan et al. (2023b). For the specific case of an analog Ising machine, we choose $q_\theta(x)$ to be a fully-factorized Bernoulli distribution over individual spins.

$$q_\theta(x) = \prod_{j=1}^{N^2} q_\theta(x_j), \quad q_\theta(x_j) = m_j^{\frac{1+x_j}{2}}(1-m_j)^{\frac{1-x_j}{2}}, \quad m_j \in [0,1].$$

where each spin generator $q_\theta(x_j)$ is a Bernoulli distribution with local magnetization $m_j$. This formulation converts the problem of generating an $N \times N$ spin configuration into generating $N^2$ local magnetizations $\{m_j\}$, which comprise the parameter vector $\theta$. For this choice, the gradient of the loss with respect to each $m_j$ can be computed analytically, see Appendix A.1 for details.

$$\nabla_{m_j} L = \mathbb{E}_{q_\theta}\Big[\beta E(x)\frac{x_j - (2m_j - 1)}{2m_j(1-m_j)} + \log\frac{m_j}{1-m_j}\Big]. \tag{6}$$

**Noisy Analog Ising Machines:** The noise in an Analog Ising machine can be observed in every measurement of the energy of a state $x$, $\tilde{E}(x)$, and can be defined as: $\tilde{E}(x) = E(x)(1 + \eta_x)$, where $E(x)$ is described by equation 1, and $\eta_x = N(0, \sigma^2); \sigma \approx 0.03(3\%)$. This noise is characterized in Appendix A.7. The BRAIN algorithm outlines a procedure for handling noisy rewards, where we find that subtracting a baseline from the measured energies is critical to reducing the variance in the gradients, and result in stable learning, see Appendix A.2 for details.

## 4 EXPERIMENTS

We systematically benchmark BRAIN against a traditional MCMC algorithm (Metropolis et al. (1953); Hastings (1970)) under noisy energy evaluations across four problems of increasing complexity and dimensionality. For a fair comparison, we assume deployment on an analog Ising machine where parallel evaluation of multiple spin states may be prohibitive, thus excluding any parallelization advantages BRAIN might have over the intrinsically sequential MCMC algorithm when evaluated on a GPU. Since evaluation of energies is the rate-limiting step, we used it as our primary metric for comparing convergence time across methods to solve each problem. Appendix A.3 discus theoretical estimates of the time to convergence. In the following sections we benchmark BRAIN against MCMC in terms of i) fidelity of solution, ii) resilience to noisy energy evaluations, iii) scalability, and iv) time-to-solution for 4 types of problems with increasing complexity in energy landscape and high-dimensional interactions.

### 4.1 BENCHMARKS WITH LOW-DIMENSIONAL ENERGY LANDSCAPES

We begin by evaluating BRAIN on a set of benchmarks with increasing complexity in their noisy energy landscapes $E(x)$, see Figure 1. The first benchmark is a standard double well energy landscape $E(x) = A(x^2 - x_0^2)^2 + Bx$, where $A, B, x_0$ are constants determining the barrier of the double-well, the offset between the two energy minima, and the location of the minima respectively (See Figure 1 (a) ) Also shown is the corresponding probability density $p(x) = e^{-\beta E(x)}/Z$, for two different inverse temperatures $\beta$. The addition of 10% noise to the energy function produces a highly irregular energy landscape away from local minima. This structural noise manifests in the probability distribution $p(x)$ as increased variability, particularly evident in the high-temperature regime. At low temperatures, however, the noise doesn't affect the likelihood of observing states significantly. Figure 1 (c) compares BRAIN and MCMC in their ability to recover $p(x)$ at both temperatures, where we find that both BRAIN and MCMC successfully recover $p(x)$ at high temperatures. However, at low temperatures, MCMC oversamples the state with lower probability, while BRAIN captures $p(x)$ correctly. The second benchmark we benchmark against is a one-dimensional six-spin case, as used in Torlai & Melko (2016b). This benchmark allows us to measure performance on an Ising system where the partition function can be evaluated exactly, allowing us to visualize and compare $p(x)$ for BRAIN and MCMC, Figure 1 (b) and (d). We find that both BRAIN and MCMC adequately sample $p(x)$ at a low temperature where two states are more likely than other states. Note that $p(x)$ is symmetric with respect to a sign inversion operation here, i.e., flipping the sign of all spins results in a state with equal $p(x)$. Therefore, when BRAIN and MCMC recover one of the two likely states,

they automatically recover the other likely state as well, via a sign inversion. The noiseless versions of these benchmarks are shown in Appendix A.5.

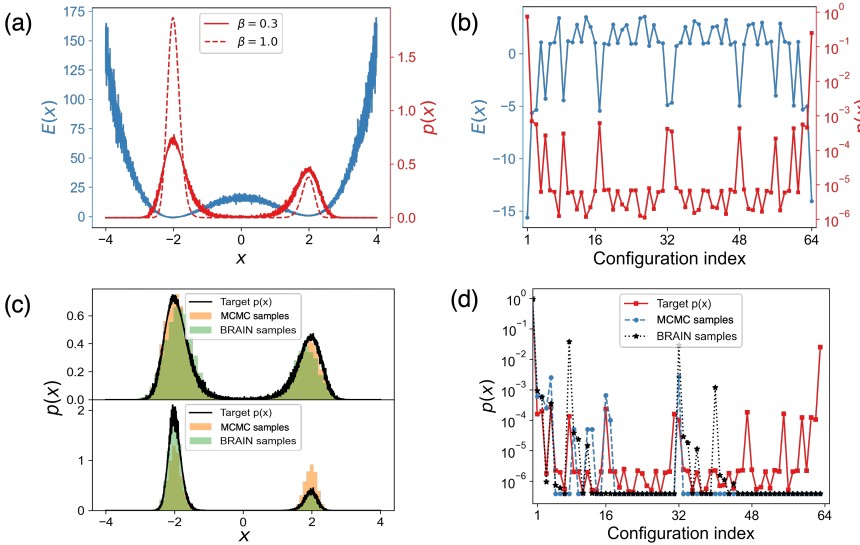

Figure 1: (a) A noisy double-well energy landscape $E(x)$, and associated probability $p(x)$ of observing state $x$. (b) The energy landscape $E(x)$ and associated probability $p(x)$ for a one-dimensional six spin system. (c) Comparing BRAIN and MCMC to the ground truth $p(x)$ for two different temperatures. We find that both MCMC and brain perform adequately at high temperature but MCMC performs worse at low temperatures (d) Comparing BRAIN and MCMC to the ground truth at a single temperature, where we find both algorithms to be equally adequate at representing $p(x)$.

## 4.2 EVALUATING BRAIN ON HIGH-DIMENSIONAL ANALOG ISING NETWORKS

### 4.2.1 NOISY CURIE-WEISS ISING HAMILTONIAN

We first evaluate BRAIN using the Curie-Weiss Hamiltonian, a canonical Hamiltonian for understanding collective magnetic phenomena and phase transitions Kochmański et al. (2013). The Curie-Weiss Hamiltonian is given by:$H = -\frac{1}{2N} \sum_{i \neq j} J_{ij} \sigma_i \sigma_j$ where $\sigma_i$, $\sigma_j$, N, and $J_{ij}(= 1)$ represents spin $i$, spin $j$, total number of spins, and the spin-spin coupling strength, respectively. This mean-field model exhibits a well-characterized second-order phase transition at the critical temperature $T_c \approx J$, making it ideal for algorithmic benchmarking. Figure 2(a) illustrates the all-to-all coupled network topology of the Curie-Weiss system, where every spin interacts with every other spin regardless of spatial distance. Such a configuration creates a homogeneous interaction landscape that eliminates geometric frustration effects.

**Noise Resilience in tracking Phase Transitions**: Under noisy conditions, the temperature-dependent magnetization profiles (Figures 2(b-c) reveal a fundamental advantage of using BRAIN over MCMC, where we see that MCMC fails catastrophically, while BRAIN shows remarkable noise resilience. In the noiseless case, see Appendix A.5, both algorithms accurately capture the thermodynamic landscape and achieve high fidelity to converge to theoretical ground state magnetization $|M| = 1$. Figure 2(b) shows that even with minimal (1%) noise, MCMC achieves only $|M| = 0.76$ instead of the theoretical $|M| = 1$ at low temperatures. This failure stems from the breakdown of the Metropolis acceptance criterion when measurement noise approaches or exceeds energy differences between competing configurations. In such condition, the acceptance probability becomes essentially random, violating the detailed balance condition necessary for Boltzmann sampling Metropolis et al. (1953); Hastings (1970). In stark contrast, Figure 2(c) shows BRAIN's remarkable noise resilience, maintaining accurate identification of $T_c$ and achieves near-optimal magnetization even with 9% noise - a performance level unprecedented for optimization algorithms operating under such severe measurement uncertainty.

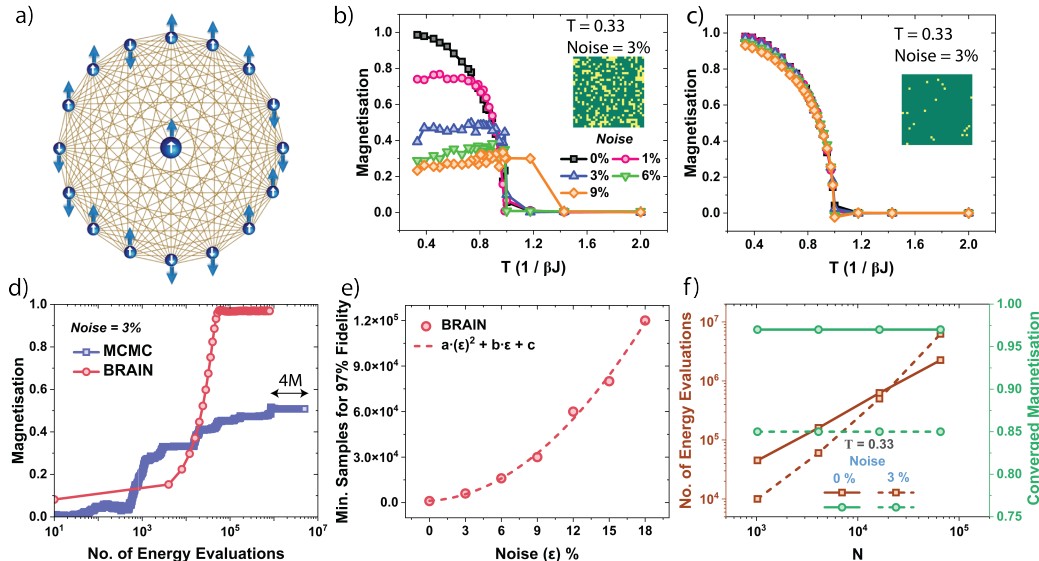

Figure 2: (a) All-to-all coupled Ising network representing the Curie-Weiss Hamiltonian where every spin on the network is coupled every other spin independent of the distance between the spins. Temperature-dependent magnetization profile for (b) MCMC and (c) BRAIN under various noise levels. The inset in (b) and (c) represents the ground-state magnetization of the spin state at T = 0.33 with 3% noise. (d) Convergence dynamics of MCMC and BRAIN at at T=0.33 under 3% noise, shown in terms of the number of energy evaluations required to reach optimal solutions. (e) Plot tracking the increase in the minimum number of samples needed for BRAIN to maintain the fidelity of the solution as the %noise increases in the analog Ising machine. (f) Scalability analysis across system sizes (N) ranging from 1,024 to 65,536 spins. The left axis displays the number of energy evaluations at T=0.33 under both noiseless and 3% noise conditions, while the right axis shows the corresponding magnetization values.

The $|M| - T$ profile generated using MCMC and BRAIN reveal that noise has the most pronounced impact at low temperatures, where both algorithms struggle to reach ground state configurations, resulting in reduced magnetization accuracy. Under realistic 3% noise conditions, representative of current photonic hardware, BRAIN achieves $|M| = 0.98$ compared to MCMC's $|M| = 0.51$, representing 97% versus 51% fidelity approaching the theoretical optimum. Corresponding spin states (green and yellow represents spin up and spin down states respectively, where the ground state solution should be all green or all yellow) at low temperature (T=0.33) are given as insets of Figure 2(b, c). In the absence of noise, both MCMC and BRAIN achieve magnetization values close to unity. However, MCMC's performance deteriorates sharply under noise, exhibiting a 75% fidelity loss in magnetization at 20% noise. Additionally, even advanced Monte Carlo methods, such as parallel tempering, perform poorly under noisy conditions (Appendix A.10). In contrast, BRAIN maintains robustness with only 13% degradation under the same conditions, see Appendix A.9.

Figure 2(d) quantitatively compares convergence behavior under realistic 3% noise conditions, revealing a critical computational advantage of BRAIN's distribution learning approach. BRAIN reaches optimal solutions within $10^5$ energy evaluations, while MCMC remains trapped in suboptimal states even after $5 \times 10^6$ evaluations with $36\times$ more energy evaluations. This comparison uses energy evaluations as the fundamental computational metric, which is particularly relevant for analog Ising systems where each energy measurement represents the rate-limiting hardware operation. We quantify the computational advantage of BRAIN over MCMC in Table 1 as a function of increasing noise in the energy evaluations.

**Scalability and Noise Tolerance Trade-offs**: Figure 2(e) reveals a critical insight into BRAIN's adaptive sampling strategy: as noise levels increase, the algorithm increases the number of samples per batch to maintain solution fidelity. We observe a quadratic dependence on the number of additional samples required by BRAIN to maintain high-fidelity solution as % Gaussian

noise ($\epsilon$) increases. Specifically we fit the second-order polynomial $y = a\epsilon^2 + b\epsilon + c$ where $a = 339.9, b = 416.7, \& c = 1190.5$. At 1% noise, BRAIN requires approximately $1.5\times$ more energy evaluations compared to noiseless conditions. This requirement grows to $2\times$ at 3% noise and $3\times$ at 6% noise, see Appendix A.6. This behavior reflects the fundamental statistical requirement: higher noise demands more samples to achieve equivalent signal-to-noise ratios in gradient estimates, following our theoretical analysis.

The scalability analysis (Figures 2(f)) demonstrates BRAIN's favorable scaling properties across system sizes from 32×32 to 256×256 (65,536 spins) while revealing important trade-offs between system size, noise tolerance, and computational requirements. Under noiseless conditions, converged magnetization remains constant across all system sizes, indicating dimension-independent performance quality. However, under 3% noise, some degradation occurs with increasing system size, with magnetization dropping to 0.85 for the largest system—representing a 15% deviation from theoretical optimum while maintaining unprecedented performance at these scales under noisy conditions. The computational scaling across system size exhibits: $63.6N^{0.94}$ for noiseless systems and $0.2N^{1.55}$ for noisy systems. The noise-induced overhead increases systematically with system size—from 25% additional evaluations at 1,024 spins to 100% at 65,536 spins—reflecting the fundamental challenge of signal discrimination in high-dimensional noisy landscapes. This scaling analysis reveals BRAIN's practical advantage for analog Ising machines: while noise increases computational requirements, the algorithm's inherent parallelizability on GPUs can offset these costs through simultaneous evaluation of hundreds or thousands of configurations, an impossibility for sequential MCMC approaches.

Table 1: Comparison of MCMC and BRAIN Methods on 1024 spin system

| | Fidelity | | Gain | Time to reach MCMC solution | | Acceleration |
|---|---|---|---|---|---|---|
| **Noise** | **MCMC** | **BRAIN** | **BRAIN/MCMC** | **MCMC** | **BRAIN** | **MCMC/BRAIN** |
| 3% | 51% | 98% | 1.9× | $1.0 \times 10^6$ | $5.2 \times 10^3$ | 192× |
| 6% | 37% | 98% | 3.3× | $1.1 \times 10^6$ | $4.5 \times 10^3$ | 244× |
| 9% | 29% | 98% | 3.9× | $1.0 \times 10^6$ | $3.5 \times 10^3$ | 285× |
| 12% | 27% | 98% | 4.5× | $0.9 \times 10^6$ | $2.4 \times 10^3$ | 408× |

### 4.2.2 LENZ–ISING MODEL: GENERALIZATION BEYOND MEAN-FIELD ISING NETWORKS

In order to demonstrate the agnostic nature of BRAIN, we evaluate performance on the 2D nearest-neighbor Ising model - a system with local interactions that serves as a benchmark for NP(non deterministic polynomial time)-hard combinatorial optimization problems where the energy of the spin-state is given by $H = -J \sum_{i,j \in N(i)} \sigma_i \sigma_j$.

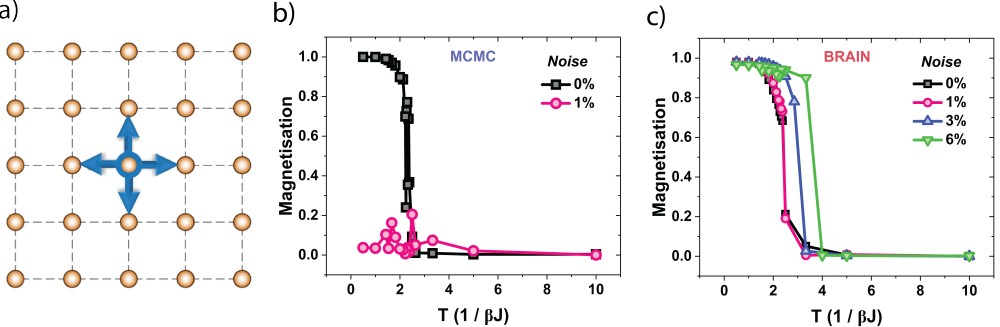

Figure 3: (a) Schematic representing a 2D square array of spins emulating the classic Ising Hamiltonian with nearest-neighbor interactions (arrows). Temperature-dependent magnetization for the 2D nearest-neighbor Ising model, comparing MCMC (b) and BRAIN (c) under increasing noise for energy evaluations.

Figure 3(a) confirms excellent agreement between BRAIN and MCMC under noiseless conditions, with both methods accurately capturing the $T_c \approx 2.2$ predicted by Onsager's exact solution Aharoni (2000). This validates BRAIN's correct reproduction of phase transition behavior in systems with local spatial correlations maintaining high-fidelity. Under noisy conditions, the algorithms again show divergent behavior. Figure 3(b) reveals MCMC's complete breakdown even at 1% noise, trapping the system in a fully disordered state ($M \approx 0$) across all temperatures - despite strong ordering expected at low T. The noise renders the Metropolis acceptance criterion essentially random, reducing sophisticated statistical mechanical sampling to random walk behavior. Figure 3(c) demonstrates BRAIN's resilience across noise levels from 0% to 6%. At all noise levels, the algorithm maintains the characteristic sigmoid magnetization-temperature curve and successfully identifies phase transitions. However, a systematic and physically meaningful trend emerges: the apparent $T_c$ shifts to higher values with increasing noise level at the rate of $\partial T_c / \partial \epsilon \approx 8.48J$. This upward shift reflects noise acting as an additional source of disorder competing with thermal fluctuations, requiring stronger thermal driving to achieve equivalent magnetic correlation—analogous to quenched disorder effects in spin glassesAizenman & Contucci (1998).

## 5 DISCUSSION AND OUTLOOK

BRAIN introduces a useful approach for optimization on analog hardware by reformulating the problem from noisy state sampling to robust distribution learning. Its core strength lies in its ability to handle noisy, non-differentiable, oracle evaluations—a good match for the characteristics of physical computing systems where energy landscapes are measured, not calculated analytically. The most notable impact is BRAIN's unprecedented performance with noise. Unlike MCMC, which fails under even low noise, BRAIN's performance degrades more slowly, and its relative advantage grows with the noise level. With 3% noise, BRAIN achieves 98% fidelity while MCMC only reaches 51%, and it finds the MCMC-level solution 192 times faster. This resilience makes it a practical solution for emerging analog hardware.

While BRAIN demonstrates several advantages over conventional MCMC, in terms of fidelity, acceleration, and resilience to noise, the current implementation has a few limitations. First, the choice of a fully-factorized Bernoulli distribution for the state generator $q_\theta(x)$ assumes spin independence and may not fully capture the long-range correlations present in the Boltzmann distributions of more complex Hamiltonian. This choice was made in part to explore simple models, and let the REINFORCE learning process, driven by an energy function that rewards correlations, drive the model to produce them. REINFORCE drives the set of independent local magnetization ($m_j$) to converge to states with highly correlated spin configurations, ensuring that state generator does not bias the learning algorithm. Future work could explore more expressive models such as normalizing flows, or graph neural networks, to generate spatially correlated spin-states at the cost of increased computational complexity. Second, our analysis is confined to a multiplicative Gaussian noise, which is representative of most hardware but may not encompass all noise sources in different analog systems. Real-world devices can exhibit non-Gaussian noise or systematic biases, or temporally correlated noise which are not explored here. Finally, the algorithm's performance relies on sufficient sampling to ensure the convergence and stability of policy gradient estimates. As shown in the Appendix A.6, too few samples can lead to suboptimal solutions, creating a practical trade-off between the acceleration and the fidelity of the learned distribution.

We demonstrated the generalization and scalability of BRAIN across multiple problems of increasing complexity in energy landscapes. Its effectiveness on both fully-connected (Curie-Weiss) and locally-connected (2D nearest-neighbor) Ising models demonstrates its model-agnostic nature. The favorable $N^{1.55}$ scaling with noisy evaluations up to 65,536 spins positions BRAIN as a viable tool for tackling large-scale, combinatorial optimization problems on next-generation hardware. Looking ahead, the BRAIN framework opens several promising research avenues. Extending the algorithm beyond classical Hamiltonian to quantum systems could enable novel methods for variational quantum algorithms. A particularly impactful application of BRAIN lies in hardware-in-the-loop co-design, where the learning algorithm adapts not to a simulated noise model but to the unique, real-time characteristics of a specific physical device. By embracing hardware imperfections, this approach may help realize the full potential of large-scale analog systems for both fundamental scientific discovery and complex optimization challenges.

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

# A APPENDIX

## A.1 GRADIENT EXPRESSION FOR THE BERNOULLI DISTRIBUTION

When $q_\theta(x)$ is the composition of independent Bernoulli distributions per spin, we can write:

$$q_\theta(x) = \prod_{j=1}^{N^2} q_\theta(x_j), \quad q_\theta(x_j) = m_j^{\frac{1+x_j}{2}}(1-m_j)^{\frac{1-x_j}{2}}, \quad m_j \in [0,1].$$

Substituting this choice of $q_\theta(x)$ into our KL-based loss, we have:

$$\begin{aligned}
L(\theta) &= -H(q_\theta(x)) + \beta\,\mathbb{E}_{q_\theta}[E(x)] \\
&= -\sum_x \Big(\prod_j q_{\theta_j}(x_j)\Big)\log\Big(\prod_j q_{\theta_j}(x_j)\Big) + \beta\,\mathbb{E}_{q_\theta}[E(x)] \\
&= \sum_j \Big[m_j\log m_j + (1-m_j)\log(1-m_j)\Big] + \beta\,\mathbb{E}_{q_\theta}[E(x)].
\end{aligned} \tag{7}$$

Given this loss function, we can efficiently learn $\theta$ using policy-gradient reinforcement learning or REINFORCE Williams (1992). This approach is particularly useful when $E(x)$ is non-differentiable or noisy, as in measurements from an analog Ising machine, where backpropagation is not applicable.

For this choice, the gradient of the loss with respect to each $m_j$ can be computed analytically as:

$$\nabla_{m_j} L = \mathbb{E}_{q_\theta}\Big[\beta E(x)\frac{x_j - (2m_j-1)}{2m_j(1-m_j)}\Big] + \log\frac{m_j}{1-m_j}. \tag{8}$$

## A.2 REDUCING THE VARIANCE IN THE GRADIENTS BY SUBTRACTING A BASELINE

**Theorem 1 (Reducing the variance in gradients with noise by subtracting a baseline):**

Let the gradient of the loss with respect to the parameters $\theta$ be defined by $g^{(s)}(\theta)$, as a function of the number of samples $s$ in a batch. Following the main manuscript, we have:

$$g^{(s)}(\theta) = \frac{1}{s}\sum_{i=1}^s \Big[\beta\tilde{E}(x_i)\nabla_\theta\log q_\theta(x_i) - \nabla_\theta H(q_\theta)\Big], \quad x_i \sim q_\theta(x),$$

where $\tilde{E}(x_i) = E(x_i)(1+\eta_i)$ includes multiplicative noise $\eta_i$ with $\text{Var}[\eta_i] = \sigma^2$. Define the baseline $b = \frac{1}{s}\sum_{j=1}^s \tilde{E}(x_j)$ and the corresponding baseline-subtracted gradient

$$g^{(s,b)}(\theta) = \frac{1}{s}\sum_{i=1}^s \Big[\beta(\tilde{E}(x_i) - b)\nabla_\theta\log q_\theta(x_i) - \nabla_\theta H(q_\theta)\Big].$$

Then the baseline-subtracted gradient has strictly smaller variance:

$$\text{Var}[g^{(s,b)}(\theta)] < \text{Var}[g^{(s)}(\theta)].$$

**Proof:** We begin by analyzing $g^{(s)}(\theta)$.

$$g^{(s)}(\theta) = \frac{1}{s}\sum_{i=1}^s \Big[\beta\tilde{E}(x_i)\nabla_\theta\log q_\theta(x_i) - \nabla_\theta H(q_\theta)\Big], \quad x_i \sim q_\theta(x)$$

Under the assumption of multiplicative noise, one can write:

$$g^{(s)}(\theta) = \frac{1}{s}\sum_{i=1}^s \beta E(x_i)\nabla_\theta\log(q_\theta(x_i)) + \frac{1}{s}\sum_{i=1}^s \beta\eta_i E(x_i)\nabla_\theta\log(q_\theta(x_i)) - \frac{1}{s}\sum_{i=1}^s \nabla_\theta H(q_\theta)$$

We have two sources of variance in this gradient, limited sampling $s$, and noise per sample $\eta_i$, which are both independent. Dropping the deterministic entropy term, we have

$$\text{Var}[g^{(s)}(\theta)] = \text{Var}[\sum_{i=1}^s \frac{\beta}{s}E(x_i)\nabla_\theta\log(q_\theta(x_i))] + \text{Var}[\sum_{i=1}^s \frac{\beta}{s}\eta_i E(x_i)\nabla_\theta\log(q_\theta(x_i))]$$

For the variance arising from sampling, we have

$$\text{Var}[\sum_{i=1}^{s} \frac{\beta}{s} E(x_i) \nabla_\theta \log(q_\theta(x_i))] = \frac{\beta^2}{s} \text{Var}[E(x) \nabla_\theta \log(q_\theta(x))]$$

For the variance arising from noise, we have

$$\text{Var}[\sum_{i=1}^{s} \frac{\beta}{s} \eta_i E(x_i) \nabla_\theta \log(q_\theta(x_i))] = \frac{\beta^2 \sigma^2}{s^2} \sum_{i=1}^{s} E(x_i)^2 (\nabla_\theta \log q_\theta(x_i))^2$$

where we use $\text{Var}[\eta_i] = \sigma$, and that the noise for each sample $i$ is independent. Putting them together, we have:

$$\text{Var}[g^{(s)}(\theta)] = \frac{\beta^2}{s} \text{Var}[E(x) \nabla_\theta \log(q_\theta(x))] + \frac{\beta^2 \sigma^2}{s^2} \sum_{i=1}^{s} E(x_i)^2 (\nabla_\theta \log q_\theta(x_i))^2$$

This variance in the gradients $g^{(s)}(\theta)$, can be reduced by using a baseline correction [see ref 25 of prl]. Let's begin by writing the gradient with the baseline

$$g^{(s,b)}(\theta) = \frac{1}{s} \sum_{i=1}^{s} \left[ \beta(\tilde{E}(x_i) - b) \nabla_\theta \log q_\theta(x_i) - \nabla_\theta H(q_\theta) \right]$$

where $b = \frac{1}{s} \sum_{j=1}^{s} \tilde{E}(x_j)$. Once again assuming multiplicative noise, we have:

$$g^{(s,b)}(\theta) = \frac{1}{s} \sum_{i=1}^{s} \beta(E(x_i) - \bar{E}) \nabla_\theta \log q_\theta(x_i)$$

$$+ \frac{1}{s} \sum_{i=1}^{s} \beta \left( \eta_i E(x_i) - \frac{1}{s} \sum_{j=1}^{s} \eta_j E(x_j) \right) \nabla_\theta \log q_\theta(x_i) - \frac{1}{s} \sum_{i=1}^{s} \nabla_\theta H(q_\theta)$$

where $\bar{E} = \frac{1}{s} \sum_{j=1}^{s} E(x_j)$. Separating the variance arising from sampling, the variance from noise, and dropping the entropy term we have:

$$\text{Var}[g^{(s,b)}(\theta)] = \text{Var}\left[ \frac{\beta}{s} \sum_{i=1}^{s} (E(x_i) - \bar{E}) \nabla_\theta \log q_\theta(x_i) \right]$$

$$+ \text{Var}\left[ \frac{\beta}{s} \sum_{i=1}^{s} \left( \eta_i E(x_i) - \frac{1}{s} \sum_{j=1}^{s} \eta_j E(x_j) \right) \nabla_\theta \log q_\theta(x_i) \right].$$

For the variance arising from the sampling, we have (analogous to before):

$$\text{Var}[\sum_{i=1}^{s} \frac{\beta}{s} (E(x_i) - \bar{E}) \nabla_\theta \log(q_\theta(x_i))] = \frac{\beta^2}{s} \text{Var}[(E(x) - \bar{E}) \nabla_\theta \log(q_\theta(x))]$$

To analyze the variance arising from noise, we introduce $a_i = \frac{\beta}{s} \nabla_\theta \log(q_\theta(x_i))$, $\delta_i = \eta_i E(x_i) - \frac{1}{s} \sum_{j=1}^{s} \eta_j E(x_j)$, such that $\Sigma_i \delta_i = 0$.

$$\text{Var}_{\text{noise}}[g^{(s,b)}(\theta)] = \text{Var}[\sum_{i=1}^{s} a_i \delta_i] = \sum_{i=1}^{s} a_i^2 \text{Var}[\delta_i] + \sum_{i \neq j} a_i a_j \text{Cov}[\delta_i, \delta_j]$$

One can then show that the variance is:

$$\text{Var}[\delta_i] = \text{Var}\left[ \eta_i E(x_i) - \frac{1}{s} \sum_{j=1}^{s} \eta_j E(x_j) \right] = \sigma^2 E(x_i)^2 (1 - \frac{2}{s}) + \frac{\sigma^2}{s^2} \sum_{k=1}^{s} E(x_k)^2$$

and the covariance is:

$$\text{Cov}[\delta_i, \delta_j] = -\frac{\sigma^2}{s} (E(x_i)^2 + E(x_j)^2) + \frac{\sigma^2}{s^2} \sum_{k=1}^{s} E(x_k)^2$$

Putting it all together, we have:

$$\text{Var}[g^{(s,b)}(\theta)] = \frac{\beta^2}{s}\,\text{Var}[(E(x) - \bar{E})\,\nabla_\theta \log q_\theta(x)]$$

$$+ \sum_{i=1}^{s} a_i^2 \left( \sigma^2 E(x_i)^2 \left(1 - \frac{2}{s}\right) + \frac{\sigma^2}{s^2} \sum_{k=1}^{s} E(x_k)^2 \right)$$

$$+ \sum_{i \neq j} a_i a_j \left( -\frac{\sigma^2}{s}(E(x_i)^2 + E(x_j)^2) + \frac{\sigma^2}{s^2} \sum_{k=1}^{s} E(x_k)^2 \right)$$

The reduction in variance is given by $\text{Var}[g^{(s)}(\theta)] - \text{Var}[g^{(s,b)}(\theta)]$. Since reducing the mean $\bar{E}$ from $E(x)$ reduces the variance, we know that the variance arising from the sampling is reduced when subtracting a baseline (although the reduction can be marginal for large $s$).

We now focus on the reduction in variance due to noise, where

$$\text{Var}_{\text{noise}}[g^{(s)}(\theta)] - \text{Var}_{\text{noise}}[g^{(s,b)}(\theta)] = \frac{\beta^2 \sigma^2}{s^2} \sum_{i=1}^{s} a_i^2 E(x_i)^2 - \sum_{i=1}^{s} a_i^2 \text{Var}[\delta_i] - \sum_{i \neq j} a_i a_j \text{Cov}[\delta_i, \delta_j]$$

plugging in expressions from above, and collecting terms:

$$\text{Var}_{\text{noise}}[g^{(s)}(\theta)] - \text{Var}_{\text{noise}}[g^{(s,b)}(\theta)] = \sum_{i=1}^{s} a_i^2 \sigma^2 \left( \frac{2}{s} E(x_i)^2 - \frac{1}{s^2} \sum_{k=1}^{s} E(x_k)^2 \right)$$

$$- \frac{2\sigma^2}{s} \sum_{i=1}^{s} a_i E(x_i)^2 \sum_{j \neq i} a_j - \frac{\sigma^2}{s^2} \sum_{k=1}^{s} E(x_k)^2 \sum_{i \neq j} a_i a_j$$

Simplifying this further results in:

$$\text{Var}_{\text{noise}}[g^{(s)}(\theta)] - \text{Var}_{\text{noise}}[g^{(s,b)}(\theta)] = \frac{2\sigma^2}{s} \sum_{i=1}^{s} (\nabla_\theta \log q_\theta(x_i))^2 E(x_i)^2 > 0$$

An order of magnitude estimate for an analog Ising machine and a Bernoulli distribution can be obtained by establishing that $E \sim N$ and $\nabla_\theta \log q_\theta(x_i)^2 \sim m_j^2 \sim \text{O}(1)$.

$$\text{Var}[g^{(s)}(\theta)] \sim \text{O}\left( \frac{\beta^2 N^2}{s} + \frac{\beta^2 \sigma^2 N^2}{s} \right) \quad \text{Var}[g^{(s,b)}(\theta)] \sim \text{O}\left( \frac{\beta^2 N^2}{s} + \frac{\beta^2 \sigma^2 N^2}{s^2} \right)$$

However, if the spins are partially correlated due to the Hamiltonian, $\text{Var}[E]$ could change from $\sim N^2$ to $\sim N^x$, making

$$\text{Var}[g^{(s)}(\theta)] \sim \text{O}\left( \frac{\beta^2 N^x}{s} + \frac{\beta^2 \sigma^2 N^x}{s} \right) \quad \text{Var}[g^{(s,b)}(\theta)] \sim \text{O}\left( \frac{\beta^2 N^x}{s} + \frac{\beta^2 \sigma^2 N^x}{s^2} \right)$$

## A.3 ESTABLISHING ORDER OF CONVERGENCE TIMES FOR BRAIN COMPARED TO MCMC

**Theorem 2:** The convergence time for BRAIN under noisy evaluations is $\tau_{\text{BRAIN}} \sim \text{O}\left( \frac{\beta^2 N^2}{s\epsilon^2} + \frac{\beta^2 \sigma^2 N^2}{s^2 \epsilon^2} \right)$ while the time to converge to a solution for MCMC, at low temperatures, is $\tau_{\text{MCMC}} \sim \text{O}\left( \frac{N}{|\beta-1|} e^{|\beta-1|^2 N} \right)$

**Proof:** A classic result from the convergence of stochastic gradient descent algorithms Garrigos & Gower (2023) states that the number of steps $\tau$ needed to reach an accuracy $\epsilon$ is given by

$$\tau \sim \text{O}(\frac{\text{Var}[g]}{\epsilon^2})$$

where we assume a constant, small learning rate. Plugging in the order of magnitude estimates from Theorem 1, we get a general estimate as:

$$\tau_{\text{BRAIN}} \sim \mathrm{O}\Big(\frac{\beta^2 N^x}{s\epsilon^2} + \frac{\beta^2 \sigma^2 N^x}{s^2 \epsilon^2}\Big)$$

where we see the utility of subtracting a baseline from the gradient, which is to reduce the affect of noisy energy measurements on the convergence time of BRAIN.

In contrast, for MCMC, a well known result Ding et al. (2009) is that the mixing time of the chain, for the analog Ising machine Hamiltonian considered in this work, is

$$\tau_{\text{MCMC}} \sim \mathrm{O}\Big(\frac{N}{|\beta-1|}e^{|\beta-1|^2 N}\Big)$$

at low temperatures. While higher temperature offer $\mathrm{O}(N \log N)$ or even $\mathrm{O}(N^{1.5})$ scaling, we are interested in capturing metastable states at all temperatures with BRAIN.

## A.4 VALIDATING BRAIN ON NOISELESS SYSTEMS

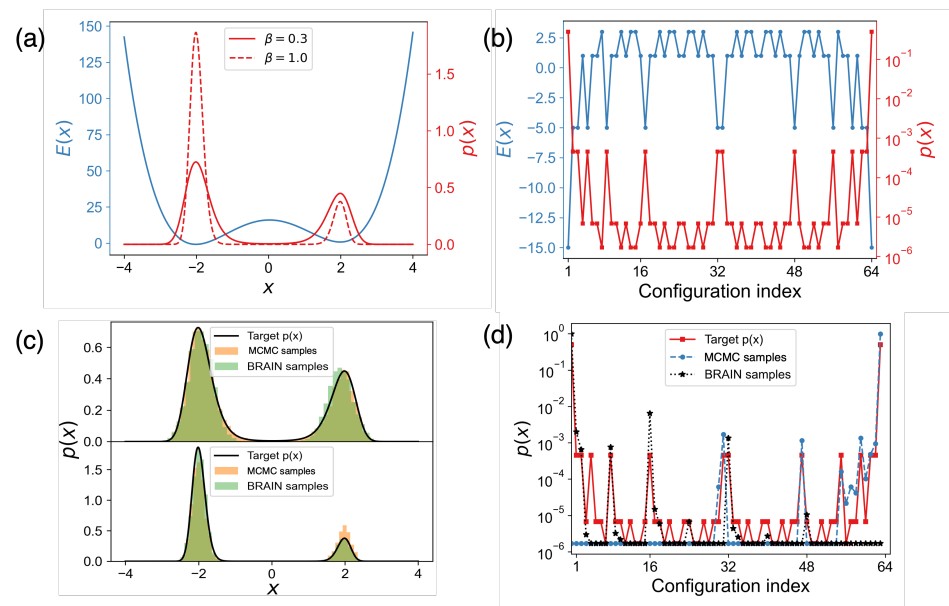

Figure 4: (a) A noiseless double-well energy landscape $E(x)$, and associated probability $p(x)$ of observing state $x$. (b) The energy landscape $E(x)$ and associated probability $p(x)$ for a one-dimensional six spin system. (c-d) Comparing BRAIN and MCMC to the ground truth $p(x)$ for two different temperatures.

Analogous to Fig. 1, Fig. 4 demonstrates the performance of BRAIN and MCMC on the test cases discussed in the main manuscript. We find that both BRAIN and MCMC are adequate at capturing $p(x)$ in the double-well energy case and the one-dimensional six-spin case.

## A.5 VALIDATING BRAIN ON NOISELESS ISING MACHINES

Figure 5(a) demonstrates demonstrates the fundamental validation of BRAIN against the established Monte Carlo Markov Chain (MCMC) method for the Curie-Weiss model on a 32×32 lattice under noiseless conditions. Both algorithms exhibit identical temperature-dependent magnetization profiles, accurately capturing the second-order phase transition at the critical temperature $T_c$sc $\approx 0.87$. The sigmoid-shaped transition curve confirms proper thermodynamic behavior, with both methods achieving complete ferromagnetic ordering ($|M| = 1$) at low temperatures and complete disorder (M $\approx 0$) at high temperatures. The excellent quantitative agreement across the entire temperature

range validates that BRAIN correctly implements the underlying statistical mechanics, reproducing the Boltzmann distribution with the same fidelity as the gold-standard Monte Carlo (MCMC) method (Figure 5(a)). This agreement is particularly significant because it demonstrates that the policy gradient approach maintains thermodynamic consistency despite using a fundamentally different optimization paradigm. The qualitative similarity between MCMC and BRAIN configurations at each temperature point confirms that BRAIN samples from the same statistical ensemble as MCMC (Figure 5(b) and 5(c)), demonstrating proper ergodic behavior and avoiding systematic biases that could lead to artificial ordering or disorder. This configurational equivalence is crucial for applications requiring accurate representation of the full Boltzmann distribution, not just ground state optimization.

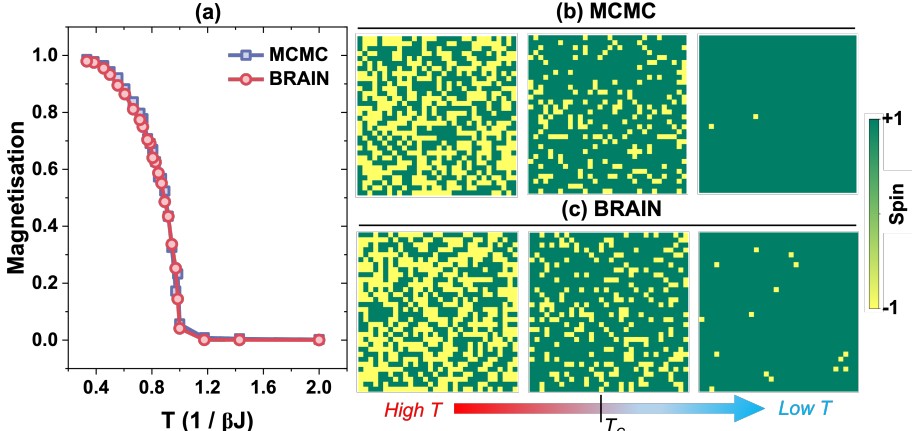

Figure 5: (a) Temperature-dependent magnetization comparison between MCMC and BRAIN for the Curie-Weiss model on a 32×32 lattice. Representative spin configurations at three temperatures: $T > T_c$ (disordered), $T \approx T_c$ (critical fluctuations), and $T < T_c$ (ordered) for (b) MCMC and (c) BRAIN.

### A.6 OPTIMIZING SAMPLE SIZE

Figure 6 investigates the relationship between sample size and optimality for the Curie-Weiss model at low temperature (T = 0.33) on a 32×32 system. For very small sample sizes (n ¡ 1,000), the algorithm shows suboptimal convergence, achieving magnetizations significantly below the theoretical optimum of $|M| = 1$. This poor performance reflects insufficient sampling for accurate gradient estimation in the BRAIN algorithm - the policy gradient estimates become too noisy to drive effective learning when based on too few samples. Then with increase in sample size, magnetization reaches the optimal value. For sample sizes, n greater than 1000, the curve plateaus near the theoretical optimum, indicating convergence to the true ground state.

### A.7 CHARACTERIZING THE NOISE IN THE ANALOG ISING MACHINE

Figure 7(a) presents direct experimental evidence of energy measurement fluctuations for a fixed spin configuration in a spatial photonic Ising machine (SPIM). Figure 7(b) shows the corresponding distribution of measured energies across multiple measurements, revealing the Gaussian noise, characteristics inherent to photonic hardware. The bell-shaped distribution confirms that measurement noise follows a normal distribution. The width of the distribution provides direct measurement of the noise standard deviation $\sigma$, which appears consistent with the 3% relative noise level used in our model.

### A.8 COMPREHENSIVE NOISE RESILIENCE ANALYSIS

Figure 8 provides a systematic investigation of BRAIN performance across an extreme range of noise levels, from perfect (noiseless) conditions to severe 40% noise that would render most algo-

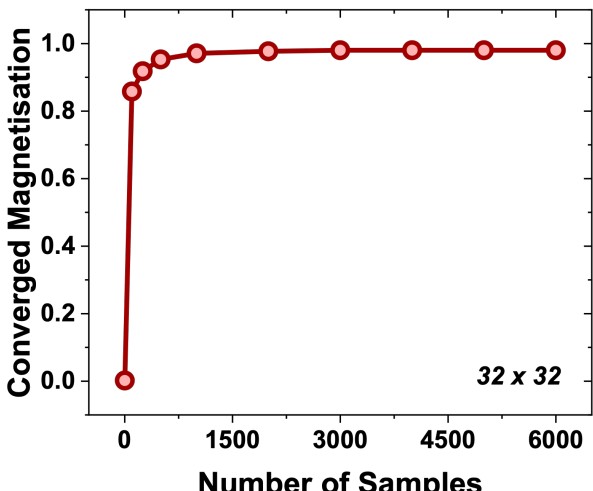

Figure 6: Effect of number of samples on the optimality of Curie-Weiss solution at T = 0.33 for a 32x32 system.

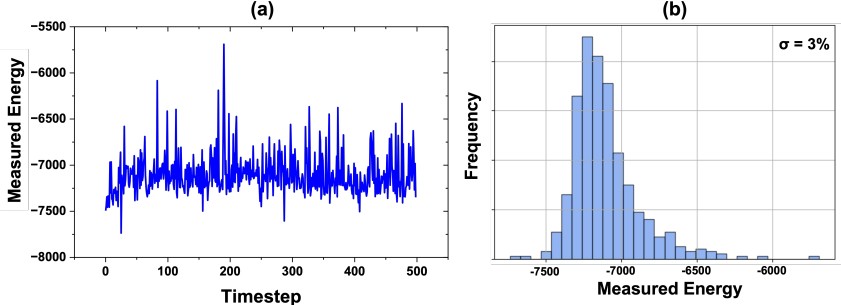

Figure 7: (a) Fluctuation in energy observed for a fixed spin configuration using the Spatial Photonic Ising Machine (SPIM) across 500 experimental trials. (b) Histogram illustrating the distribution of measured energy values.

rithms completely ineffective. This comprehensive analysis, conducted within the framework of the Curie-Weiss Hamiltonian, reveals the remarkable robustness of the BRAIN under conditions that far exceed realistic hardware specifications. At low noise levels (0-9%), the algorithm maintains near-perfect performance, with magnetization-temperature curves virtually indistinguishable from the noiseless case. The critical temperature remains accurately identified, and the transition sharpness is preserved, indicating that moderate noise levels pose minimal challenge to the algorithm. Even under severe noise conditions (10-40%), the algorithm continues to exhibit recognizable phase transition behavior, albeit with increased transition broadening and reduced low-temperature magnetization. The persistence of ordered phases under such extreme conditions demonstrates the fundamental robustness of BRAIN.

### A.9 COMPARATIVE ALGORITHM PERFORMANCE UNDER NOISE

Figure 9 provides a direct quantitative comparison between MCMC and BRAIN as a function of noise level in Curie-Weiss Hamiltonian at low temperature (T = 0.33). The contrasting behavior of the two algorithms reveals fundamental differences in their noise handling capabilities. The MCMC curve shows catastrophic degradation even at minimal noise levels. At just 1% noise, the magnetiza-

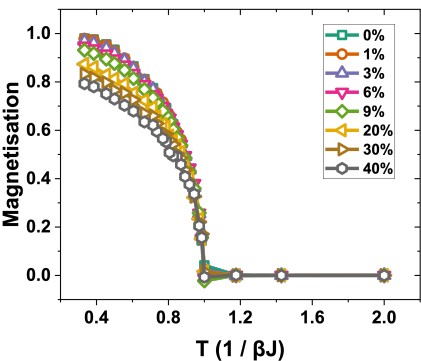

Figure 8: (a) Temperature-dependent magnetization profile for BRAIN under various noise levels ranging from 0 to 40% for a 32x32 system

tion drops quickly from the theoretical optimum of M = 1.0 to approximately M = 0.5, representing a 50% loss in optimal solution. Further increases in noise level cause continued degradation, with the algorithm essentially failing to distinguish between random and ordered configurations at higher noise levels. This failure stems from the fundamental mechanics of the Metropolis acceptance criterion, which relies on accurate energy difference calculations ($\Delta E$). When noise magnitude approaches or exceeds the energy differences between competing configurations, the acceptance probability becomes essentially randomized, violating the detailed balance condition necessary for proper Boltzmann sampling. In stark contrast, the BRAIN maintains robust performance across the

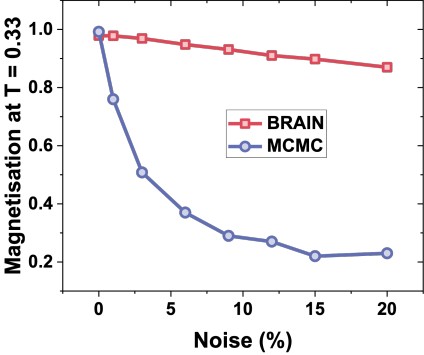

Figure 9: Comparing converged magnetization at low temperature (T=0.33) as a function of noise for MCMC and BRAIN.

entire noise range. Even at 20% noise—a level that would be considered extreme in any practical implementation—BRAIN achieves magnetization values above 0.8, representing less than 20% degradation from the optimal solution. The shallow slope of BRAIN's performance curve indicates that the algorithm's noise resilience scales gracefully with noise level, rather than exhibiting the abrupt failure characteristic of MCMC methods. This behavior reflects the algorithm's ability to aggregate information across multiple samples and adapt to noise-corrupted energy landscapes through gradient-based learning.

### A.10    Parallel tempering fails with Noise on the Curie-Weiss Hamiltonian

We implemented parallel tempering Earl & Deem (2005) as an advanced Monte Carlo baseline to investigate whether sophisticated sampling enhancements could overcome noise limitations in com-

puting the Curie-Weiss Hamiltonian. The algorithm employed 30 replicas spanning temperatures from 0.33 to 2, with replica exchange attempts every 4,000 Monte Carlo steps over 400,000 total time steps per replica. While the method performed well under noiseless conditions, achieving efficient sampling across the temperature range with successful configuration exchanges between adjacent replicas (Figure 10), it failed completely under 3% noise conditions. All replicas remained trapped in disordered states ($M \approx 0$) despite the extended simulation time and frequent swap opportunities, with successful exchange events becoming extremely rare due to corrupted energy evaluations. This systematic failure across multiple temperature levels and extended sampling demonstrates that the fundamental limitation is not specific to simple Monte Carlo, but extends to advanced variants that still rely on single-measurement decision criteria incompatible with noisy energy evaluations. The contrast between noiseless success and noisy failure (Figure 10) confirms that measurement uncertainty, rather than algorithmic sophistication, represents the critical bottleneck for traditional statistical mechanical approaches on analog computing platforms.

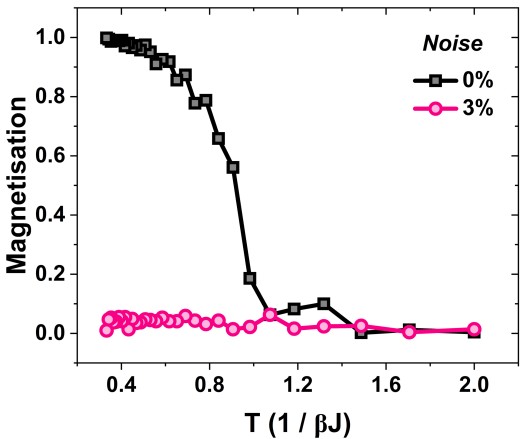

Figure 10: Magnetization as a function of temperature for the parallel tempering algorithm, comparing performance under noiseless (0% noise) and noisy (3%) energy evaluations.

