# OpenReview forum: "BRAIN: Boltzmann Reinforcement For Analog Ising Networks"
_ICLR.cc/2026/Conference — ICLR 2026 Conference Withdrawn Submission_

### Official Review · Reviewer_swjF · 2025-10-17

**Soundness:** 1
**Presentation:** 2
**Contribution:** 1
**Rating:** 0
**Confidence:** 3

**Summary:**

This paper introduces a method to approximate a Boltzmann distribution by minimizing the reverse KL divergence between a simple product distribution and the target distribution. The approach is evaluated on the Ising model, with the authors claiming superior performance over MCMC methods when noisy energy functions are observed.

**Strengths:**

- The paper addresses a challenging problem in statistical physics and combinatorial optimization, which is relevant to both theoretical and applied research.

**Weaknesses:**

- **Novelty:** The primary contribution appears to be the use of noisy observations of the energy value. However, the core methodology does not introduce substantial conceptual or technical advancements beyond existing literature.
- **Model Expressivity:** The choice of a simple product distribution as the approximating model is concerning. Such distributions are inherently limited in expressivity and typically struggle to capture multimodal or complex target distributions [6]. This raises questions about the validity of results like those in Figure 1c, where the model seemingly approximates a bimodal distribution—a task that should theoretically exceed the capabilities of a product distribution.
- **Literature Awareness:** The paper does not adequately engage with recent, relevant work in the field, such as:
  - [6] Wu et al. (2019) and [7] Nicoli et al. (2020) who consider an autoregressive distributio on the Ising Model
  - [1] Karalias and Loukas (2020), which explores neural frameworks for combinatorial optimization on graphs.
  - [2] Sanokowski et al. (2024), which introduces a diffusion model framework for unsupervised neural combinatorial optimization.
  - [3] Sanokowski et al. (2025), focusing on scaling discrete diffusion samplers and evaluates diffusion samplers in sampling settings
  - [4] Holderriethet al. (2025) and [5] Ou et al. (2025), who propose discrete neural flow samplers with locally equivariant transformers and evaluate on the Ising model
- **Technical Accuracy:** There are problematic statements in the paper, such as the claim in line 217 that a graph neural network serves as a model distribution. This is misleading, as GNNs are used as a parametrization of model distributions.

**Questions:**

1. **Novelty:** How does this work differentiate itself from the approaches outlined in [1]–[5]? Could you clarify the specific technical or conceptual innovations introduced in this paper?
2. **Model Capability:** How can a simple product distribution, which is theoretically limited to unimodal distributions, produce samples that approximate a bimodal distribution, as shown in Figure 1c? Are there additional components or assumptions not clearly stated in the paper?
3. **Literature Engagement:** Why were the works [1]–[5] not discussed or compared against in the paper? How does your method relate to or improve upon these existing approaches?

---

[1] Karalias, Nikolaos, and Andreas Loukas. "Erdos goes neural: an unsupervised learning framework for combinatorial optimization on graphs." Advances in Neural Information Processing Systems 33 (2020): 6659-6672.

[2] Sanokowski, Sebastian, Sepp Hochreiter, and Sebastian Lehner. "A Diffusion Model Framework for Unsupervised Neural Combinatorial Optimization." International Conference on Machine Learning. PMLR, 2024.

[3] Sanokowski, Sebastian, et al. "Scalable Discrete Diffusion Samplers: Combinatorial Optimization and Statistical Physics.

[4] "Holderrieth, Peter, Michael Samuel Albergo, and Tommi Jaakkola. "LEAPS: A discrete neural sampler via locally equivariant networks." Forty-second International Conference on Machine Learning.

[5] Ou, Zijing, Ruixiang Zhang, and Yingzhen Li. "Discrete Neural Flow Samplers with Locally Equivariant Transformer." arXiv preprint arXiv:2505.17741 (2025).

[6] Wu, Dian, Lei Wang, and Pan Zhang. "Solving statistical mechanics using variational autoregressive networks." Physical review letters 122.8 (2019): 080602.

[7] Nicoli, Kim A., et al. "Asymptotically unbiased estimation of physical observables with neural samplers." Physical Review E 101.2 (2020): 023304.

---

### Official Review · Reviewer_ZEoy · 2025-10-24

**Soundness:** 2
**Presentation:** 1
**Contribution:** 1
**Rating:** 2
**Confidence:** 5

**Summary:**

This method employs the REINFORCE gradient estimator to learn an approximation of the Boltzmann distribution of Ising systems based on noisy energy measurement from an Analog Ising machines. The method uses the reverse KL divergence as an objective function and compares the sampling performance primarily against Metropolis Hastings MCMC that also uses the noisy energy measurements.

**Strengths:**

- Clear problem motivation and good motivation for using analog hardware in these problems.

**Weaknesses:**

- The paper does not sufficiently explain the role of the analog Ising machine clearly enough. Please explain what it is, what it does, and how it is used.

- No comparison to any other ML/RL method: this paper only evaluates REINFORCE gradient estimator in conjunction with a variational distribution that is a Bernoulli product distribution. Please employ also state-of-the-art RL methods like PPO and please also consider more sophisticated parameterizations of the variational distribution like e.g. discrete diffusion models (Sanokowski et al. 2025). The chosen approach cannot be considered state-of-the-art and consequently also lacks novelty from an ML perspective.

- Compare against methods that are not based on analog Ising machines (see Sanokowski et al. 2025 and references therein).

Sanokowski, S., Berghammer, W., Ennemoser, M., Wang, H. P., Hochreiter, S., & Lehner, S. (2025). Scalable discrete diffusion samplers: Combinatorial optimization and statistical physics. arXiv preprint arXiv:2502.08696.

**Questions:**

- Why is MH chosen as the primary MCMC benchmark and not more sophisticated MCMC methods like Sequential MC or Parallel Tempering (which is only mentioned once in the appendix)?

---

### Official Review · Reviewer_HR86 · 2025-11-01

**Soundness:** 2
**Presentation:** 3
**Contribution:** 2
**Rating:** 4
**Confidence:** 3

**Summary:**

This work proposes an optimization algorithm for analog Ising networks. Instead of sampling individual states and comparing them to determine the optimization direction, the proposed method directly learns to approximate the full Boltzmann distribution. Compared to MCMC, the experimental results demonstrate stronger noise resilience and better scalability to large systems.

**Strengths:**

- The manuscript is clearly structured.

- The proposed method exhibits strong robustness to measurement noise.

**Weaknesses:**

- The comparison is somewhat limited. Only MCMC is used as a baseline, while more advanced techniques (e.g., neural Boltzmann generators) are not considered.

- The experimental section lacks ablation studies, such as analyses of different choices of $q_{\theta}$ or different noise models.

**Questions:**

As a non-expert in this subfield, I find it difficult to distinguish the key differences between the proposed method and a variational Boltzmann machine. Could the authors clarify this distinction more explicitly?

---

### Note · Authors · 2025-11-28

**Comment:**

We would like to thank the reviewers for their insightful comments. Based on reviewer feedback, we would like to withdraw the paper and take some time to address the reviewer's comments to strengthen our paper, and more clearly state our contribution on algorithms that handle noisy oracles (energy evaluators).

**Withdrawal Confirmation:**

I have read and agree with the venue's withdrawal policy on behalf of myself and my co-authors.